# Occupational exposure to HIV among nurses at a major tertiary hospital: Reporting and utilization of post-exposure prophylaxis; A cross-sectional study in the Western Cape, South Africa

Katlego Tebogo Kabotho[1], Tawanda Chivese[2]*

1 Division of Epidemiology and Biostatistics, Department of Global Health, Faculty of Medicine and Health Sciences, University of Stellenbosch, Cape Town, South Africa, 2 Department of Population Medicine, College of Medicine, QU Health, Qatar University, Doha, Qatar

* tchivese@qu.edu.qa

**Data Availability Statement:** All relevant data are within the manuscript and its Supporting Information files.

## Abstract

### Background

While treatment for HIV has greatly improved patient outcomes, health care workers, including nurses, remain at high risk of occupational exposure. The risk of exposure is a continuous concern in the South African health system that is overburdened by multiple stressors, including the highest HIV caseload in the world. The aim of this study was to estimate the prevalence of occupational exposure to HIV, reporting and utilization of post-exposure prophylaxis, knowledge, attitudes towards HIV post-exposure prophylaxis and infection control practices amongst nurses at a tertiary hospital in the Western Cape, South Africa.

### Methods

A cross-sectional study was conducted at Tygerberg hospital from the 4th to the 16th February 2019. Participants were front line nurses working in randomly selected wards. A self-administered questionnaire was used to collect data from participants.

### Results

Of the 160 participants who took part in the survey, 17 reported occupational exposure to HIV (prevalence 10.6%, 95% CI 6.7–16.6), and of the 17 exposed, 10(58.8%) reported needlestick injuries. From those who were exposed, only 10 (58.8%) reported the incidents and went on post-exposure prophylaxis. However, only 6 out of the 10 completed their treatment. Half (50%) of the participants had inadequate knowledge on HIV post-exposure prophylaxis, 83.3% had adequate attitudes towards HIV post-exposure prophylaxis and 75% had adequate infection control practices.

**Funding:** The authors received no specific funding for this work.

**Competing interests:** The authors have declared that no competing interests exist.

**Abbreviations:** CDC, Centres for Disease Control; CI, Confidence interval; HCW, Health care workers; HIV, Human immunodeficiency virus; NSI, Needlestick injury; PEP, Post-exposure prophylaxis; PI, Percutaneous injuries; WHO, World Health Organization.

## Conclusion

One out of every nine nurses had occupational exposure to HIV at a major tertiary hospital with poor reporting and utilization of post-exposure prophylaxis. The high proportion of needle stick injuries highlights the need for better infection control training. Similarly, the low levels of HIV post-exposure prophylaxis knowledge show the need for structured intervention and in-service training for health care workers.

## Background

HIV infections continue to be on the rise worldwide despite different strategies and policies that have been implemented over the years. Recent statistics show that globally, 37.9 million people are currently living with HIV while 1.7 million were recently infected with HIV during the year 2018[1]. Due to the rise in HIV infections, there is increasing contact of healthcare workers (HCW) with people living with HIV. HCWs are at risk of exposure to HIV infected material during their work, especially in sub-Saharan Africa, where the majority of people living with HIV reside. Globally, South Africa has the highest number of people living with HIV, currently estimated at 7.7 million [1]. Those at a higher risk of exposure to HIV, including HCWs must be equipped with knowledge in order to prevent infection [2].

The World Health Organization (WHO) provides recommendations that guide health care workers in the event of exposure to HIV in the workplace [3]. In South Africa, post-exposure prophylaxis (PEP) is widely available especially for HCWs [4]. Nonetheless, HCWs must have adequate information about HIV PEP as this will inform their action post-exposure. Evidence from several studies has shown that the overall level of knowledge regarding HIV PEP is generally inadequate among healthcare workers [5, 6, 7]. This is despite reported positive perceptions regarding the treatment [8]. Determinates for poor uptake of HIV PEP by those exposed are not well known, although these may include poor knowledge, poor attitudes and lack of clear guidelines and reporting pathways [9].

In South Africa, knowledge and practices to protect HCWs from blood-borne diseases remain inadequate in low resource settings [10]. This may be due to lack of training and inadequate exposure to relevant information on HIV PEP. In Cameroon, 80 nurses were surveyed to assess their knowledge and practices on HIV PEP. In all, 73.7% of the participants had poor knowledge about HIV PEP [11]. Although the majority in the previously mentioned study (83.8%), had heard about PEP, only 10 (12.5%) received formal training on PEP. This shows a gap in HIV PEP training in low-resource facilities.

Reviewed literature has shown that nurses are the most affected by occupational exposure to HIV compared to other cadres in the health sector. In 2016, a study in Ethiopia indicated nurses, apart from medical doctors, were the most affected by exposures with 58.2%, compared to 30.8% laboratory and 23.3% other professions[12]. In another survey in South West Ethiopia, higher rates of percutaneous injuries were observed among midwives and nurses (91.7% and 81%), compared to doctors 77.8% [13].

In 2017, a systematic review and meta-analysis with 65 cross-sectional studies from 21 African countries, reported a high pooled lifetime and 12-month prevalence of occupational exposure to body fluids of 65.7% and 48.0%, respectively [14]. Exposure was largely due to percutaneous injury with an estimated 12-month prevalence of 36.0% (95% CI: 31.2–40.8). In 2016, a study in Botswana revealed that 26% of participants had been exposed to sharps injuries or splashed with fluids 3 months prior to the survey [15]. Of these, only 160 (37%)

reported the exposure to the relevant persons and 67% of these went on to take HIV PEP treatment, however, only 71% of them completed their medication.

In South Africa, a few studies have investigated exposure to HIV among HCWs, although data is lacking from the major tertiary hospitals. During 2008, 42 out of 53 (79.2%) intern doctors reported exposure to blood or body fluids of which 64% were percutaneous injuries and 36% mucosal [16]. Similarly, in 2019 at the Far East Rand Hospital in the Gauteng Province, 136 out of 175 (77.7%) reported exposure to blood and body fluids among interning doctors [17]. To our knowledge, data on nurses' exposure to HIV at the workplace in South Africa are scarce. We found only one study that investigated knowledge and uptake of PEP amongst nurses caring for people living with HIV in Limpopo [18]. Findings from the aforementioned study revealed that approximately 40% of nurses working that the hospital did not know what PEP was, and 22% did not know or were not sure if it was available in the hospital [18]. In the Western Cape, the prevalence of occupational exposure to HIV among HCWs is not known, as there are no recently published studies.

The main objective of this study was to estimate the prevalence of occupational HIV exposure, reporting and utilization of PEP in nurses at a major referral hospital in the Western Cape of South Africa. Further, the research aimed to estimate the nurses' knowledge and attitudes towards HIV PEP as well as assess their practices towards infection control. A secondary aim of the research was to investigate associations between demographic and professional characteristics of the nurses and risk of occupational exposure to HIV.

## Methods

### Study design and setting

A cross-sectional study was conducted at Tygerberg Hospital. This is the largest hospital in the Western Cape and the second largest hospital in South Africa with an estimated number of eight thousand staff members and about two thousand nurses. Tygerberg hospital has more than 1384 beds and offers 28 specialist services [19].

### Study population

The study population consisted of nurses working at the facility, recruited from the 4th to the 16th of February 2019. All consenting nurses who worked directly with patients were included in the study. Nurses in management and administrative positions were excluded from taking part in the study as their work does not usually involve direct contact with patients or possibly infectious material.

### Statistical considerations

**Sample size determination.** The sample size was determined using an acceptable margin of error of 7%, 95% confidence level and a single population proportion formula considering a prevalence of 60%, most studies reported a prevalence between 40–80%. Studies from Ethiopia [12] and Nigeria [20] reported a prevalence rate of 46% and 67.5% respectively, while in South Africa two studies [15, 16] revealed a prevalence of 79.2% and 77.7% respectively.

The total required sample size was calculated using the formula below:

$$N = \frac{1.96^2 \, x \, (1-p)^2}{d^2}$$

• where p = 0.6, d = 0.07

The overall estimated sample size was therefore 126. To ensure we reached the required sample size while taking into consideration possible 20% non-response (due to declines, absenteeism, and lack of time for participation) a sample size of 152 participants was required.

**Sampling technique.** A two-stage cluster random sampling using *Stata 15 statistical software* [21] was used to select participants for the survey. Ten departments were randomly selected out of an overall 24 departments, followed by random selection of 2 wards from each of the 10 departments. All the nurses in the selected wards were invited to participate in the study.

## Data collection

A self-administered questionnaire was used to collect data (S1 Doc). The questionnaire was developed after reviewing the literature [8, 18]. A pilot study was done to test the procedure and to determine the relevance of questions on 20 interning medical students from the same hospital. For the attitude section, the calculated Cronbach alpha was 0.54.

We made a few amendments to each section of the questionnaire after the pilot study. In section 1 of demographics, we changed the option, "caucasian" to "white" in the question "which race do you identify yourself as?" Section 2 was about occupational exposure to HIV, in question 2a we added the instruction "if the answer is no, please proceed forward to section 3 of the questionnaire" we also added question 2g, "did you start on treatment?". A few questions were dropped in section 3 (which was about knowledge on HIV PEP) because of their repetitive nature in section 3. The question, "have you attended any training on HIV PEP?" was removed because it appeared in section 1. Question 3f, "under which circumstances would you not take HIV PEP?" was also removed because it was similar to 3d. Some questions were amended to add options from which the participants were to choose their answers, for instance, the option, "I don't know", was added to section 3 and 4 (attitudes towards HIV PEP) of the questionnaires for those who did not know the answer to the question. In section 5, which is about HIV infection control practices, we dropped the question, "what would prompt you to take PEP?", because it was similar to a question asked in section 3.

We obtained written informed consent from the participants, and after this, the questionnaires were distributed to all the nurses in the ward at the time. To minimize response bias due to night staff not participating, questionnaires for the night staff were left with the nurse in charge for them to complete. These questionnaires were collected the next morning.

## Data extraction and statistical analysis

Data were captured into *Microsoft Office Excel* and exported to *STATA 2015* for analysis.

Where numerical data were normally distributed, means and standard deviations were used to summarize the data. Where numerical data were not normally distributed, medians and interquartile ranges (IQR) were used to summarize the data. For categorical data, frequencies and percentages were reported.

**Determination of occupational HIV exposure.** The overall prevalence of occupational HIV exposure was calculated from the number of self- reported incidents relative to the total number of study participants. The 95% confidence interval (95%CI) of the prevalence was also reported.

**Determination of knowledge.** Four questions were asked to assess the knowledge of the participants on HIV PEP. The overall knowledge score was computed by adding up all the correct knowledge answers and divided by the total number of questions asked, then expressed as a percentage. A median (IQR) knowledge percentage score was reported. The percentage score

was categorised as "poor" if a participant scored <50%, "good" if a participant scored between 50–74% and "adequate if a participant scored ≥75%.

**Determination of attitudes.** Attitudes were determined using a 5-point Likert scale. Responses ranged from strongly agree to strongly disagree. However, there was an option for those who did not know the answer to the question to state so. All the participants who selected "I don't know" were not included in the analysis therefore, the responses "don't know" were treated as missing data. To get the overall attitudes mark, all 6 questions were scored individually, for questions where the preferred responses were "strongly agree/agree" or "strongly disagree/disagree" a mark of 1 was given and a zero was given if they chose a different answer from the preferred one. We then added all the individual scores and expressed them as percentages. To summarize the attitude scores, median and IQR were reported. Further, attitude scores were categorised as "poor" if a participant scored <50%, "good" if a participant scored between 50–74% and "adequate if a participant scored ≥75%, as done in previous studies[22]. Frequencies and percentages for each question were also presented.

**Determination of practices.** Four questions were asked to assess the practices of participants regarding infection control. The overall practice score was computed by adding up all the correct practices per question and divided by the total and then expressed as a percentage. A median (IQR) practice percentage score was reported. The percentage score was categorised as "poor" if a participant scored <50%, "good" if a participant scored between 50–74% and "adequate if a participant scored ≥75% [22].

## Comparisons and hypothesis testing

We compared demographic data, and median (IQR) scores for knowledge, attitudes and practices between participants who were exposed to HIV and those not exposed. The chi-squared test or Fisher exact test (where chi-squared was not valid) were used to compare categorical variables between exposed and unexposed participants. The Wilcoxon rank-sum test was used to compare knowledge, attitude and practice scores between exposed and unexposed participants. A p-value of 0.05 was considered statistically significant.

**Investigation of factors associated with exposure.** A multiple variable logistic regression was used to investigate the determinants of occupational exposure to HIV. Occupational exposure to HIV was treated as a binary outcome while the predictor variables were the practice score, professional training, experience and demographic variables. The 95%CI were reported for odds ratios.

## Ethics

The study was carried out according to the principles of the Helsinki Declaration [19]. Ethical clearance was obtained from the medical research ethics committee in Stellenbosch University (reference number 7751) while permission to carry out the study was received from the Tygerberg Hospital research committee and Western Cape Province (reference number WC_201809_012).

## Results

### Sociodemographic characteristics of participants

A total of 168 nurses were approached to take part in the study and, of these, 160 agreed to participate. Eight participants did not consent to take part in the study, resulting in a 95.24% response rate. One participant did not complete the demographic section of the questionnaire. Most of the participants, 147 (92.45%) were female. The mean age (SD) of the participants was

**Table 1. Socio-demographics of participants, compared between participants with occupational HIV exposure and those not exposed.**

| Variable | | Overall | Exposed, N = 17 | Not exposed, N = 143 | P value |
|---|---|---|---|---|---|
| Gender | Females, n (%) | 147 (92.45) | 14 (9.52) | 133 (90.48) | 0.099 |
| | Males, n (%) | 12 (7.55) | 3 (25) | 9 (75) | |
| Race | Black, n (%) | 73 (45.91) | 3 (4.11) | 70 (95.89) | 0.013 |
| | White, n (%) | 4 (2.52) | 0 | 4 (100) | |
| | Mixed ancestry, n (%) | 81 (50.94) | 14 (17.28) | 67 (82.72) | |
| Marital status | Single, n (%) | 68 (43.40) | 7 (10.29) | 61 (89.71) | 0.902 |
| | Married, n (%) | 70 (44.03) | 8 (10.43) | 62 (88.57) | |
| | Divorced, n (%) | 15 (9.43) | 1 (6.67) | 14 (93.33) | |
| | Widowed, n (%) | 5 (3.14) | 1 (20) | 4 (80) | |
| Age | Mean (SD) | 40.63(9.91) | 45(34–49) | 41 (34–48) | 0.563 |
| Years of practice | Years | 9 (4–24) | 15 (6–25) | 8.5(4–24) | 0.254 |
| Education, n (%) | diploma | 126 (79.25) | 12 (9.52) | 113 (90.48) | 0.433 |
| | bachelor | 20 (12.58) | 4 (20) | 16 (80) | |
| | Masters | 1 (0.63) | 0 | 1 (100) | |
| | other | 12 (7.55) | 1 (8.33) | 11 (91.67) | |
| Training attendance on HIV PEP, n (%) | yes | 40 (25.16) | 3 (7.5) | 37 (92.5) | 0.443 |
| Departments/wards | Medical n (%) | 80(50.31) | 5(6.25) | 75(93.75) | 0.199 |
| | Casualty n (%) | 39(24.53) | 6(15.38) | 33(84.62) | |
| | OPD n (%) | 40(25.16) | 6(15.00) | 34(85.00) | |

40.63 years (SD 9.91). Only 40 (25.16%) of the 160 participants had attended formal training on HIV PEP. The departments were categorised into 3 major groups; medical, casualty and OPD. The department that had most respondents was the medical 80(50.31) followed by OPD 39(24.53) and casualty 40(25.16). A summary of the demographics by exposure status is shown in Table 1.

## Frequency and reporting of occupational exposure to HIV, and PEP utilization

Amongst the 160 participants who took part in the study, 17 of the respondents got occupationally exposed to HIV in the past 12 months of their work, resulting in an overall prevalence of 10.63% (95% CI 6.72–16.60%). From the 17 who were exposed, only 10 (58.82%) reported the incidents and sought treatment (Fig 1). All the 10 participants who reported exposures started on treatment. However, only 6 (60.0%) of the 10 respondents completed treatment resulting in a 40.0% dropout rate. Out of the 17 who were exposed, 10 (58.82%) had needle stick injuries, 1 (5.88%) had a cut by a sharp object while 6 (35.29%) had contact with body fluids as shown in Fig 2. Of the three participants who discontinued treatment, 2 reported that it was due to the side effects of the medication while the other person assumed that it was enough and stopped before the treatment was complete. Out of 160 participants that took part in the study, 22 (13.75%) completed the questionnaires at home or during night duty.

## Knowledge of participants about HIV PEP

Out of 159 respondents with complete data, 115 (72.33%) of the participants responded that they had heard about HIV PEP. From these 115 respondents, 57 (49.6%) named training as the source of awareness. Out of 115 participants, 70(44.03%) of the participants had knowledge that PEP should not be administered if the patient they were exposed to is HIV negative. In

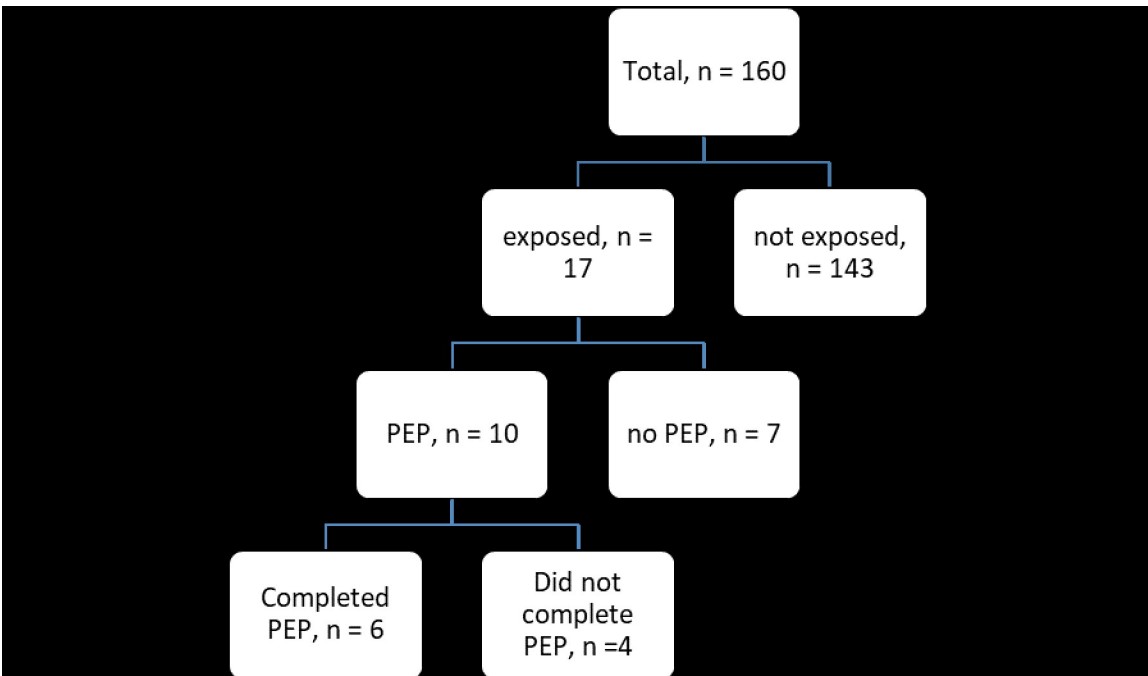

**Fig 1. Occupational exposure to HIV, reporting and utilisation of PEP.**

question three, 93 (58.49%) answered correctly about the recommended time to take PEP and in question four 66 (41.51%) answered correctly about the length of time to take PEP (Table 2). The overall median knowledge score was 50% (IQR 25–75). When the knowledge score was categorised, only 46.54% had "adequate" knowledge while 32.08% had "poor" knowledge of HIV PEP. There were no significant differences, between participants who had occupational exposure to HIV and those not exposed, in either the knowledge percent score or categorized knowledge percent score (Table 3).

## Attitudes of participants towards HIV PEP

For question 1, participants were asked for about the importance of PEP and 95.2% thought it is important. In question 2, all of the participants strongly agreed that training on HIV PEP is

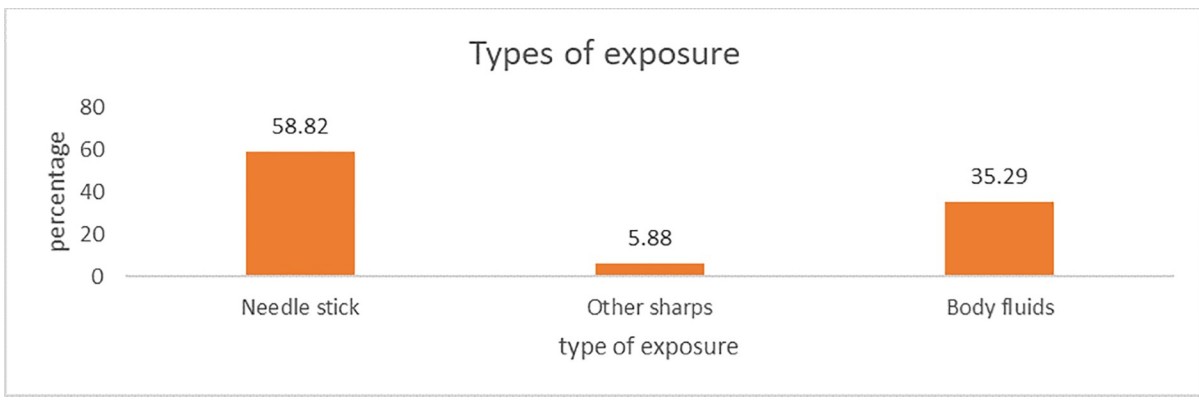

**Fig 2. Occupational exposure to HIV classification.**

**Table 2. Knowledge regarding HIV PEP.**

| Question | Response | N (%) |
|---|---|---|
| Ever heard about PEP | Yes | 115 (72.33) |
| Situation when PEP should not be administered | When the source is HIV negative? | 70 (44.03) |
| | When the patient is known to be HIV positive | 20(12.58) |
| | When HIV status of source is unknown | 16 (10.06) |
| | I don't know | 52 (32.70) |
| Recommended time to take PEP | Any time after exposure, does not matter when | 36 (22.64) |
| | Within 72 hours of exposure | 93 (58.49) |
| | I don't know | 30(18.87) |
| Length of time to take PEP | For 28 days | 66 (41.51) |
| | For six months | 28(17.61) |
| | For a lifetime | 6 (3.77) |
| | I don't know | 59 (37.11) |

important to influence people to comply with PEP guidelines. In the third question, 99.3% of the participants agreed or strongly agreed that that HIV PEP guideline poster should be posted on the walls of their working area. In question four, 85.11% of the participants agreed or strongly agreed that PEP reduces the likelihood of being HIV positive after exposure. In question 5, only 10.14% of the participants agreed or strongly agreed that HIV PEP should be administrated if a patient was HIV negative or of unknown status. In question six, 89.86% agreed or strongly agreed that PEP should be indicated for any type of sharp injuries during contact with patients of unknown HIV status (Table 4). Most participants had adequate attitudes towards HIV PEP, with a median score of 83.33% (IQR 66.67–83.33). There were no significant differences, between participants who had occupational exposure to HIV and those not exposed, in either the attitude percent score or categorized attitude percent score (Table 3).

**Table 3. Overall scores and categories of knowledge, attitudes and practices, compared between participants with occupational exposure to HIV and those not exposed.**

| | | Overall | Exposed to HIV, n = 17 | Unexposed to HIV, n = 143 | p-value |
|---|---|---|---|---|---|
| Knowledge score percent | Median (IQR) | 50 (25–75) | 75 (50–75) | 50 (25–75) | 0.258 |
| Knowledge score categories, n (%) | Poor | 51 (32.08) | 3 (17.65) | 48 (33.80) | 0.258 |
| | Good | 34 (21.38) | 3 (17.65) | 31 (21.83) | |
| | Adequate | 74 (46.54) | 11 (64.71) | 63 (44.37) | |
| Attitude score percent | Median (IQR) | 83.33 (66.67–88.33) | 83.33 (66.67–88.33) | 83.33 (66.67–88.33) | 0.373 |
| Attitude score categories, n (%) | Poor | 1 (0.67) | 1 (0.75) | 0 | 0.928 |
| | Good | 49 (32.89) | 5(31.25) | 88 (66.17) | |
| | Adequate | 99 (66.44) | 11(68.75) | 88 (66.17) | |
| Practice score Percent | Median IQR | 75 (75–100) | 100 (75–100) | 75 (75–100) | 0.153 |
| Practice score categories, n (%) | Poor | 5 (3.18) | 1 (5.88) | 4 (2.86) | 0.496 |
| | Good | 22 (14.01) | 1 (5.88) | 21 (15.00) | |
| | Adequate | 130 (82.80) | 15 (88.24) | 115 (82.14) | |

1. Exposed to HIV denotes participants that have been exposed to HIV occupationally

2. The percentage scores for knowledge, attitudes and practices were categorised as <50% poor, 50–74% good and ≥75% as adequate practice

**Table 4. Attitudes of participants towards HIV PEP.**

| Response | Strongly agree | Agree | Neutral | Disagree | Strongly disagree |
|---|---|---|---|---|---|
| Importance of PEP n (%) | 115(7.23) | 26(17.69) | 4(2.72) | 1(0.68) | 1 (0.68) |
| Perception of compliance as a result of training n (%) | 108(72.48) | 41(27.52) | 0 | 0 | 0 |
| Perception of HIV PEP guideline posters on the walls in the n (%) | 113(75.33) | 36(24.00) | 1(0.67) | 0 | 0 |
| Perception of PEP reducing likelihood of testing HIV positive after n (%) | 70(49.65) | 51(36.17) | 11(7.80) | 9 (6.38) | 0 |
| Perception of PEP treatment for exposure from HIV negative patients n (%) | 64(46.38) | 42(31.16) | 17(12.32) | 14(9.42) | 1 (0.72) |
| Perception of PEP treatment for any sharps injury for patients of unknown HIV status n (%) | 79(57.25) | 45(32.61) | 12(8.70) | 0 | 2(1.45) |

## Practices of participants regarding infection control

From a total of 159 respondents, 151 (94.97%) reported that they used personal protective equipment (PPE) whenever contact with patient blood and body fluids was anticipated. Out of 156 participants, 154 (98.72%) reported that they always discarded sharps in an appropriate container. One hundred and seventeen out of 153 (76.47%) participants reported that they never recap needles. From 153 participants who responded to question 3 "when do you seal the sharps disposal bin?" 93(60.78%) practice the right exercise (Table 5). The median practice score was 75% (IQR 75–100).

When the practice score was categorised, 82.8% of the participants had "adequate" while 3.18% had "poor" infection control practices. There were no significant differences, between participants who had occupational exposure to HIV and those not exposed, in either the practice percent score 100% (IQR 75–100) vs 75% (IQR 75–100), p = 0.153 or categorized knowledge percent score (Table 3).

## Exploration of factors associated with occupational exposure to HIV

After multiple variable logistic regression (Fig 3), there was a trend towards lower risk of exposure in female nurses, compared to males (OR 0.27, 95%CI 0.05–1.37, p = 0.114) and a trend towards higher risk in nurses untrained in HIV PEP, compared to those who received formal training in PEP (OR 2.97 95% CI 0.64–13.83 p = 0.166), although not statistically significant. There were no significant associations between occupational exposure to HIV and any of the other demographic characteristics as well as infection control practices scores.

**Table 5. Practices towards infection control measures.**

| Question | Answer | N (%) |
|---|---|---|
| Do you use personal protective equipment when anticipating contact with patient blood and body fluid?, n(%) | Yes | 151 (94.97) |
| Under what circumstances do you dispose needled and sharp objects into the dedicated biohazard bins?, n(%) | Every time | 154 (98.72) |
| | When used on an HIV positive patient | 2 (1.28) |
| Do you recap needles?, n(%) | Yes, if not used on a patient | 36 (23.53) |
| | Never | 117 (76.47) |
| When do you seal the sharps disposal bin?, n(%) | When ¾ full | 93 (60.78) |
| | When half full | 14 (9.15) |
| | When completely full | 46 (30.07) |

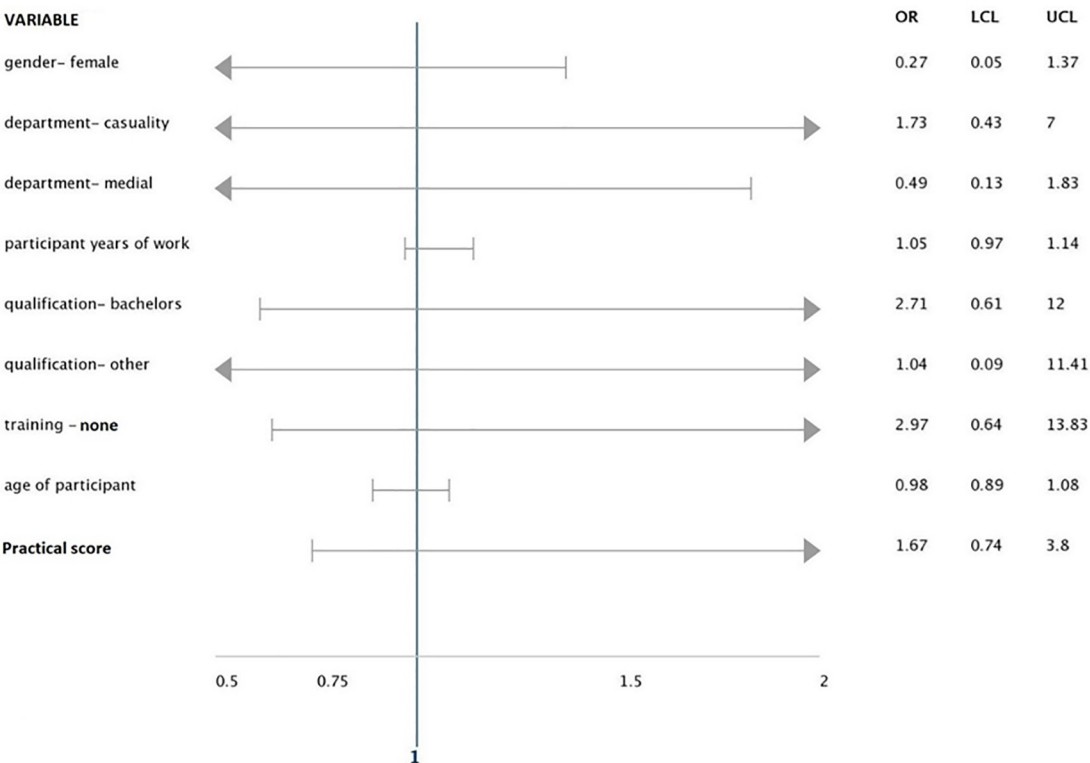

**Fig 3. Multiple variable logistic regression for factors associated with occupational exposure to HIV.** LCL- lower confidence level (lower 95%CI). UCL- upper confidence level (Upper 95%CI). OR- odds ratio.

## Discussion

In this study, we found that one out of every nine nurses at a major tertiary hospital in the Western Cape of South Africa had occupational exposure to HIV, of which almost two-thirds were exposure due to needle stick injuries. We also found inadequate reporting of exposures to occupational health services and poor PEP completion rate. The findings show that half of the participants had inadequate knowledge of HIV PEP, although most of the participants had both adequate attitudes towards PEP and good practices towards infection control.

The current exposure rate (11%) in this study is one of the lowest compared to other literature. A survey amongst HCWs in Ethiopia [12] and Nigeria [20] reported a prevalence rate of 46% and 67.5% respectively, while in South Africa two studies revealed a prevalence of 79.2% and 77.7% respectively [15, 16] in a cohort of medical interns. The prevalence rate close to the one reported in this study is 19.2%, from a study carried out amongst HCWs in Uganda [21]. In this study, 65% of the exposures were attributable to percutaneous injuries with most of them due to needle stick injuries (58.8%). In the year 2015 in Tanzania, a report showed a 62.9% exposure to needle stick injuries [5]. Similarly, an exposure rate of 63.6% was reported from a study done in Nigeria in 2011 [20]. In most studies needle stick injuries account for most of the exposures and our findings follow the same trend.

Only 59% of the participants reported exposure incidents. There were several reasons. A lack of clear knowledge of what to do after the exposure was frequently the case in our study. One participant indicated that after exposure she did not report the case to the relevant authorities as she was told by a fellow colleague that she would have to go on HIV PEP treatment for 6 months. Another participant revealed that a colleague of hers' told her "not to worry about

the exposure" as it was "just a blood splash" so she did not report it. In South-East Ethiopia, 59% of participants did not report injuries due to the following reasons; time constraints, sharps which caused injury were not used on any patient, the source patients did not have the disease of concern, and lack of knowledge that it should be reported [6]. This may have serious implications as undocumented exposure injuries could prevent injured HCWs from receiving PEP therefore potentially resulting in new HIV infections. Another concern is that 40% of participants did not finish PEP. Our findings are comparable to those of a study done Botswana in 2016, where 69% of participants received PEP and only 71% completed their medication [15]. In this study, two participants discontinued treatment due to side effects of medication. Although our study is small, and the exposed numbers were also few, the lack of completion of PEP is a clinical and public health problem which warrants attention. Larger studies may be needed to explore this and to find out the reasons why HCWs are not completing PEP, despite advancement in ARV formulations, which are now less toxic.

In this study, 50% of the participants had inadequate knowledge regarding HIV PEP. These findings are consistent with data from similar studies in Africa. In the year 2015, Cameroon had 74% of the participants with inadequate knowledge about HIV PEP[11] whereas 37% was reported in North-East Ethiopia. The overall attitudes of the participants towards HIV PEP were positive with a median score of 83% similar to findings from Botswana in 2019 [23], where participants had a mean score of 82.2% positive attitudes towards PEP. The participants had adequate practices of 75% overall. However, one of the respondents' who experienced a needle stick injury reported being pricked by a needle that was wrapped in a small opaque plastic bag as she was cleaning the area. This shows improvement is still required in infection control practice. The finding that 30% of respondents seal the sharps container when full, is a concerning result as a sharps container should never be filled up to the "full" mark. One participant mentioned that often when it's filled to the top and cannot close, they physically remove the needled or force the bin to close. The potential for injury and HIV infection from these actions cannot be understated. There seems to be a gap in infection control training and more training may be required.

One limitation of the study was that we relied on subjective recall of participants, which may have resulted in either underreporting or over-reporting of exposures. In addition, participants were allowed to take the questionnaire home if they needed more time, we also left questionnaires for night staff to complete and this could have led to participants consulting if they were unsure of the correct answers. However, only a small proportion, 22(13.75%), were given the questionnaires to take home or to do during night duty. The KAP section could have been improved by adding more questions. Only 40 participants received training on PEP while 57 heard about PEP from overall, implying that the 17 heard about PEP from other sources. Another limitation of the study is that clustering was not considered in the analysis plan, as the study was mainly explorative. Due to the relatively small sample size, we did not distinguish between the different levels of nurses, although this may have had an effect on the risk of exposure. The Cronbach alpha for the attitude scores of 0.54 was low but could be partly explained by the fact that the total number of questions for attitudes were only 7. More questions and more rigorous construction of the questions on attitude may help improve the internal consistency of this section. Lastly, our study was not powered up to detect factors associated with risk of exposure as it was a minor objective. Future studies, with larger sample sizes, are needed to further explore our findings.

## Conclusion

One out of every nine nurses is exposed to occupational HIV, with a worrisome proportion due to needle stick injuries, amid poor reporting of exposures and poor utilization of PEP.

Despite overall acceptable attitudes towards PEP and good practices in infection control, half of the nurses at this tertiary level do not have good knowledge of HIV PEP.

## Supporting information

**S1 Doc.**
(PDF)

**S1 Dataset.**
(CSV)

## Acknowledgments

We acknowledge all the nurses that took part in the survey as well Mr Bukasa who is a nurse's manager in orthopaedic surgery ward in Tygerberg hospital for helping out in the data collection process. We also acknowledge Ms Lindiwe J. Modise for assistance with proofreading.

## Author Contributions

**Conceptualization:** Katlego Tebogo Kabotho, Tawanda Chivese.

**Data curation:** Katlego Tebogo Kabotho.

**Formal analysis:** Katlego Tebogo Kabotho, Tawanda Chivese.

**Funding acquisition:** Katlego Tebogo Kabotho.

**Investigation:** Katlego Tebogo Kabotho.

**Methodology:** Katlego Tebogo Kabotho, Tawanda Chivese.

**Project administration:** Katlego Tebogo Kabotho.

**Resources:** Katlego Tebogo Kabotho.

**Software:** Katlego Tebogo Kabotho, Tawanda Chivese.

**Supervision:** Katlego Tebogo Kabotho, Tawanda Chivese.

**Validation:** Katlego Tebogo Kabotho.

**Visualization:** Katlego Tebogo Kabotho.

**Writing – original draft:** Katlego Tebogo Kabotho, Tawanda Chivese.

**Writing – review & editing:** Tawanda Chivese.

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
