## [Decision Letter · Decision Letter 0]

27 Nov 2019

PONE-D-19-25121

Occupational exposure to HIV among nurses at a major tertiary hospital: Reporting and utilization of post exposure prophylaxis; A cross-sectional study in the Western Cape, South Africa

PLOS ONE

Dear Ms Kabotho,

Thank you for submitting your manuscript to PLOS ONE. After careful consideration, we feel that it has merit but does not fully meet PLOS ONE’s publication criteria as it currently stands. Therefore, we invite you to submit a revised version of the manuscript that addresses the points raised during the review process.

ACADEMIC EDITOR: The reviews have recommended revisions to your manuscript. Please consider these comments especially regarding further details of the methodology and improvements to the presentation of the results.

We would appreciate receiving your revised manuscript by Jan 11 2020 11:59PM. To enhance the reproducibility of your results, we recommend that if applicable you deposit your laboratory protocols in protocols.io, where a protocol can be assigned its own identifier (DOI) such that it can be cited independently in the future. For instructions see: http://journals.plos.org/plosone/s/submission-guidelines#loc-laboratory-protocols

We look forward to receiving your revised manuscript.

Kind regards,

Tanya Doherty, PhD

Academic Editor

PLOS ONE

Journal Requirements:

2. Please include additional information regarding the survey or questionnaire used in the study and ensure that you have provided sufficient details that others could replicate the analyses. For instance, if you developed a questionnaire as part of this study and it is not under a copyright more restrictive than CC-BY, please include a copy, in both the original language and English, as Supporting Information. Additionally, please state when participants were recruited for this study.

3. Please provide additional details regarding participant consent. In the ethics statement in the Methods and online submission information, please ensure that you have specified what type of consent you obtained (for instance, written or verbal, and if verbal, how it was documented and witnessed).

4. Please remove ‘I would like to thank the Lord, my heavenly Father for the grace, wisdom, strength and courage He has afforded me to complete this thesis’ from the Acknowledgements section of this manuscript.

Reviewers' comments:

Reviewer's Responses to Questions

**Comments to the Author**

1. Is the manuscript technically sound, and do the data support the conclusions?

Reviewer #1: Yes

Reviewer #2: Yes

2. Has the statistical analysis been performed appropriately and rigorously? 

Reviewer #1: I Don't Know

Reviewer #2: Yes

3. Have the authors made all data underlying the findings in their manuscript fully available?

Reviewer #1: No

Reviewer #2: Yes

4. Is the manuscript presented in an intelligible fashion and written in standard English?

Reviewer #1: No

Reviewer #2: Yes

5. Review Comments to the Author

Reviewer #1: Background. Line 86-87 “To our knowledge, there are no recent studies published on nurses’exposure to HIV at the workplace in South Africa.” Perhaps include the following study done in Limpopo: Makhado, L., Davhana- Maselesele, M. Knowledge and uptake of occupational post-exposure prophylaxis amongst nurses caring for people living with HIV. Curationis. 2016, 39(1), a1593. http://dx.doi.org/10.4102/ curationis.v39i1.1593

Sampling and analysis: Was clustering considered in the analysis strategy (regression model)?

Questionnaire validity reliability: Describe the questionnaire validity and reliability, for example, was content validity of the questionnaire checked by experts in the field? Was a Cronbach alpha calculated for the attitudes scale?

Data collection: How reliable is the knowledge score if participants did not complete this under supervision? Indicate the date of data collection (months and year).

Results: Education of nurses – did the authors distinguish between the different levels of nurses, for example assistant nurses (auxiliaries), enrolled nurses and professional nurses?

Results: In lines 236 & 250, question numbers are mentioned, but is not clear to which questions they refer. Perhaps the authors could indicate the question numbers in Tables 2 and 4 or perhaps just refer to the table to prevent repetition.

General: The tables should be formatted consistently, for example, the font type. Figures are not very clear on the manuscript draft.

Acknowledgements: Consult the journal guidelines for what is required in the acknowledgements section.

References: Formatting should be attended to. Some references do not appear to be correct, e.g. line 384. Journal names, volumes, issue, pages, date, doi, etc. to be added consistently.

Reviewer #2: In general a well written manuscript. One concern is the validity and reliability of the questionnaire used - suggest that this is added to the manuscript. Additional comments have been made on the attached document for consideration.

6. PLOS authors have the option to publish the peer review history of their article (what does this mean?). If published, this will include your full peer review and any attached files.

Reviewer #1: Yes: Talitha Crowley

Reviewer #2: No

---

## [Author Response · Author response to Decision Letter 0]

17 Feb 2020

1.Background. Line 86-87 “To our knowledge, there are no recent studies published on nurses’ exposure to HIV at the workplace in South Africa.” Perhaps include the following study done in Limpopo: Makhado, L., Davhana- Maselesele, M. Knowledge and uptake of occupational post-exposure prophylaxis amongst nurses caring for people living with HIV. Curationis. 2016, 39(1), a1593. http://dx.doi.org/10.4102/ curationis.v39i1.1593

Thank you for your feedback. We have cited this study in the Background. 

Background line 88-92

2.Sampling and analysis: Was clustering considered in the analysis strategy (regression model)?

Thank you for your feedback. Clustering was not included in the analysis as the study was mainly exploratory. We have included this limitation in the Discussion section 

Discussion section line 371 - 372

3.Data collection: How reliable is the knowledge score if participants did not complete this under supervision? Indicate the date of data collection (months and year). 

Thank you for your feedback. We have added under the discussion, the limitations due to the fact that some participants were given the questionnaire to complete at home and not under supervision.

The data were collected over a period of two weeks from the 4th to the 16th February 2019. This has been added to the methods 

Methods section line 160-161

Discusion section line 372-374

4.Questionnaire validity reliability: Describe the questionnaire validity and reliability, for example, was content validity of the questionnaire checked by experts in the field? Was a Cronbach alpha calculated for the attitudes scale?

Thank you for your feedback. The questionnaire was adopted from existing studies, however some questions that were added and some removed, and a pilot study was done. We did not calculate a Cronbach alpha for the attitudes scale.

After the comment from the reviewer, we calculated the Cronbach alpha, which was 0.54. This has been added to the methods and as a limitation to the discussion. 

Methods section line 136-139

Method section line 139

Discussion section, lines 374-377

5.Results: Education of nurses – did the authors distinguish between the different levels of nurses, for example assistant nurses (auxiliaries), enrolled nurses and professional nurses?

Thank you for your feedback. There was no distinguishing between levels of nurses in the study. We have added this as a limitation under the Discussion section.

Discussion section line 371-372

6.Please provide additional details regarding participant consent. In the ethics statement in the Methods and online submission information, please ensure that you have specified what type of consent you obtained (for instance, written or verbal, and if verbal, how it was documented and witnessed).

Thank you for your feedback. We obtained written consent from the participants. This has been added in the methods section and on the online submission system.

Methods section line 153-154

7.Please include additional information regarding the survey or questionnaire used in the study and ensure that you have provided sufficient details that others could replicate the analyses. For instance, if you developed a questionnaire as part of this study and it is not under a copyright more restrictive than CC-BY, please include a copy, in both the original language and English, as Supporting Information. Additionally, please state when participants were recruited for this study.

Thank you for your feedback, we have included the questionnaire as part of additional information. It is in English. We have also stated when participants were recruited into the study.

Methods section lines 109-110 

8. Please remove ‘I would like to thank the Lord, my heavenly Father for the grace, wisdom, strength and courage He has afforded me to complete this thesis’ from the Acknowledgements section of this manuscript 

Thank you for the feedback, this statement has been removed from the acknowledgements section 

Acknowledgements section line 386-389

9. Data Availability statement, you have not specified where the minimal data set underlying the results described in your manuscript can be found. PLOS defines a study's minimal data set as the underlying data used to reach the conclusions drawn in the manuscript and any additional data required to replicate the reported study findings in their entirety. All PLOS journals require that the minimal data set be made fully available. For more information about our data policy, please see http://journals.plos.org/plosone/s/data-availability.

Thank you for your feedback. The data set is available and uploaded as a supplementary comma delimited data set

---

## [Editor Report · Decision Letter 1]

21 Feb 2020

Occupational exposure to HIV among nurses at a major tertiary hospital: Reporting and utilization of post exposure prophylaxis; A cross-sectional study in the Western Cape, South Africa

PONE-D-19-25121R1

Dear Dr. Kabotho,

We are pleased to inform you that your manuscript has been judged scientifically suitable for publication and will be formally accepted for publication once it complies with all outstanding technical requirements.

With kind regards,

Tanya Doherty, PhD

Academic Editor

PLOS ONE
---

## [Editor Report · Acceptance letter]

6 Mar 2020

PONE-D-19-25121R1 

Occupational exposure to HIV among nurses at a major tertiary hospital: Reporting and utilization of post-exposure prophylaxis; A cross-sectional study in the Western Cape, South Africa 

Dear Dr. Kabotho:

I am pleased to inform you that your manuscript has been deemed suitable for publication in PLOS ONE. Congratulations! Your manuscript is now with our production department. 

With kind regards,

on behalf of

Professor Tanya Doherty 

Academic Editor

PLOS ONE